# Associations between Maternal Nutritional Status, Hemodynamic Parameters, and Delivery Outcomes in Low-Risk Pregnancies: A Prospective Observational Study

**DOI:** 10.3390/nu16020183

**Published:** 2024-01-05

**Authors:** Chiara Lubrano, Francesca Parisi, Chiara Coco, Elisabetta Marelli, Eleonora Burello, Irene Cetin

**Affiliations:** 1Department of Biomedical and Clinical Sciences, Università degli Studi di Milano, 20157 Milan, Italy; chiara.lubrano@unimi.it (C.L.); chiara.coco@unimi.it (C.C.); elisabetta.marelli@unimi.it (E.M.); eleonora.burello@unimi.it (E.B.); irene.cetin@unimi.it (I.C.); 2Department of Woman, Mother and Child, Luigi Sacco and Vittore Buzzi Children Hospitals, ASST Fatebenefratelli-Sacco, Università degli Studi di Milano, 20154 Milan, Italy; 3Department of Mother, Child and Neonate, Fondazione IRCCS Ca’ Granda Ospedale Maggiore Policlinico, 20122 Milan, Italy

**Keywords:** hemodynamic, pregnancy, USCOM, healthy diet, maternal nutrition, obesity

## Abstract

Maternal nutritional status represents a pivotal predictor of pregnancy outcome. This prospective observational study investigates the associations between maternal characteristics and nutritional habits at term, hemodynamic parameters, and pregnancy outcomes. Healthy women with singleton uncomplicated pregnancies were enrolled at 36–41 gestational weeks. At enrollment, a nutritional score (0–10) was calculated in order to quantify maternal adherence to a healthy diet and lifestyle. Maternal hemodynamic parameters were assessed by using the Ultrasonic Cardiac Output Monitor (USCOM), including cardiac output (CO), systemic vascular resistance (SVR) and Smith–Madigan inotropy index (SMII). Pregnancy outcomes were recorded at delivery. Associations between maternal characteristics and nutritional score, hemodynamic parameters, and pregnancy outcomes were investigated by using multi-adjusted generalized linear models. In total, 143 pregnancies were enrolled. Pregestational body mass index (BMI) was positively associated with SVR, and negatively associated with CO and SMII. Additionally, a positive association was detected between the nutritional score and SMII. Finally, CO was positively associated with birth and placental weight, while RVS showed a negative association with birth and placental weight. This study shows that maternal derangements in nutritional status and habits are associated with a compromised hemodynamic profile at term, with additional impacts on intrauterine growth.

## 1. Introduction

Cardiovascular adaptations represent the most relevant physiological changes of the maternal body during pregnancy, ensuring adequate utero-placental blood perfusion for intrauterine growth and development. Maternal hemodynamic changes are already detectable in the first weeks of pregnancy and include increased heart rate (HR), cardiac output (CO), stroke volume (SV) and Smith–Madigan inotropy index (SMII), as well as decreased mean arterial pressure (MAP) and systemic vascular resistance (SVR). Both systolic blood pressure (SBP) and diastolic blood pressure (DBP) decrease during pregnancy with SBP decreasing slightly less compared to DBP, due to increased arterial compliance [1,2]. In this setting, the Ultrasonic Cardiac Output Monitor (USCOM) plays a pivotal role in investigating maternal hemodynamics in a non-invasive, reproducible, and time-saving way by assessing the flow through the aortic valve with a non-imaging continuous-wave Doppler transducer [3,4,5,6]. Several validation studies were performed in both non-pregnant [6,7,8,9] and pregnant populations, thus providing normal reference values of SV, CO, and SVR in uncomplicated singleton pregnancies [4,10,11]. Nevertheless, the widespread use of USCOM-1A is related to the study of hemodynamic maladaptations among complicated pregnancies. Indeed, increased SVR and decreased CO characterize several adverse outcomes, including fetal growth restriction (FGR) [12,13,14,15], hypertensive disorders [16,17], preterm premature rupture of the membranes (pPROM) [18], and gestational diabetes (GDM) [19]. Additionally, Vinayagam et al. observed that overweight and obese pregnant women showed lower cardiac output index (CI, cardiac output adjusted for body surface area) and higher systemic vascular resistance index (SVRI, systemic vascular resistance adjusted for body surface area), MAP, and HR compared to lean controls, thus laying the foundation for the association between maternal nutritional status, hemodynamic parameters, and adverse outcomes in pregnancy [20]. In obese pregnant women, a deranged hormonal and/or inflammatory milieu is likely to contribute to increased risks of adverse outcomes (i.e., GDM, hypertensive disorders, and preterm delivery). In fact, the adipose tissue acts as a source of pro-inflammatory mediators (Tumor necrosis factor-alpha (TNF-a), IL-1, IL-6, insulin, and leptin) [21] resulting in endothelial damage and systemic low-grade chronic inflammation [22,23]. Additionally, changes in plasma progesterone and estradiol concentrations are related to the hemodynamic derangements in obese pregnant women [24,25,26,27].

Conversely, several studies have demonstrated that maternal healthy diet and lifestyle contribute to improving short- and long-term reproductive and pregnancy outcomes [28,29,30]. The theory of fetal programming suggests that external environmental stimuli (i.e., smoke or alcohol habits, maternal stress, depression, or diet) can impact intrauterine life through epigenetic mechanisms, influencing the development and physiological metabolic functions of the fetus and placenta [31]. Therefore, maternal adherence to a Western-type dietary pattern has been strongly associated with increased risks of adverse pregnancy outcomes, as well as non-communicable disease in adulthood. Indeed, reduced micronutrient intake during fetal life, followed by excessive caloric intake during postnatal life, appears to be a major determinant of increased chronic-degenerative diseases (cardiovascular, metabolic, and renal). Additionally, an unhealthy diet and lifestyle are negatively associated with the modulation of inflammatory and oxidative pathways and hormonal milieu [30,32,33,34], leading to endothelial damage and alterations in placental development, and therefore an increased risk of adverse outcomes.

Therefore, it is plausible to think that maternal nutritional habits, independently of obesity, may influence pregnancy outcomes through the modulation of maternal hemodynamic adaptations.

The aim of this study is to investigate the associations between maternal characteristics, nutritional habits, hemodynamic parameters, and delivery outcomes among low-risk term pregnancies.

## 2. Materials and Methods

### 2.1. Study Population and Recruitment

This was a prospective observational study including all singleton uncomplicated pregnancies delivering at Vittore Buzzi Children Hospital, Milan, Italy, between May 2023 and August 2023.

The study was conducted in accordance with the Declaration of Helsinki and in compliance with all current Good Clinical Practice guidelines, local laws, regulations, and organizations. The protocol was approved by the Medical Ethical and Institutional Review Board. All participants gave their informed consent to collect personal data (Institutional Review Board ‘Comitato Etico Milano Area 1’ Prot. code n° 46091, date of approval 24 October 2018).

Eligible patients had uncomplicated singleton pregnancies between 36 and 41 gestational weeks acceding to routine antenatal care, spontaneous conception, or assisted reproductive technology (ART) pregnancy. Exclusion criteria were multiple pregnancy, maternal age under 18 years, language barrier, any chronic comorbidity or therapy, and any pregnancy complication (i.e., GDM, pPROM, FGR, and hypertensive disorders of pregnancy) or a known fetal abnormality. At enrollment, all participants filled out a general questionnaire detailing demographic, obstetric, and clinical data (age, educational level, employment and marital status, smoke habits, and obstetric and medical history). Anthropometric measures were recorded (height, pregestational weight and body mass index (BMI), and gestational weight gain (GWG)). An adapted version of the nutritional checklist developed and based on the International Federation of Gynecology and Obstetrics (FIGO) recommendations on adolescent, preconception, and maternal nutrition was used at enrollment [35,36]. The FIGO nutritional checklist consists of ten questions with a binary yes/no answer investigating the consumption frequency of meat (two to three times per week), fruit and vegetables (at least five portions per day), fish (once to twice weekly), dairy products (daily), whole grains (at least once per day), sweets and snacks (less than five times per week), folic acid supplementation (at least during the pre-conceptional period and the first trimester), the use of iodized salt, sun exposure (at least 10 to 15 min per day), and hemoglobin concentrations (higher than 110 g/L). Every affirmative answer provided 1 point, finally counting a score of 0 (low adherence) to 10 (high adherence) measuring adherence to a healthy diet and lifestyle in pregnancy [30]. The hemodynamic assessment was performed by the same trained researcher (CL) by using the USCOM under standardized conditions (maternal left lateral decubitus position to avoid aortocaval compression of the uterus). USCOM is a non-invasive device that employs advanced Doppler ultrasound to monitor blood flow through the aortic valve with a non-imaging probe in the suprasternal notch. Maternal hemodynamic parameters including BSA (body surface area), HR, SV, CO, MAP, SVR, and SMII were recorded [18]. SMII, an inotropism index describing contractility modulation, denotes the heart’s ability to adjust its contractile strength in response to stimuli. It quantifies the total kinetic and potential cardiac energy expenditure during systole. Additionally, longitudinal ultrasound fetal (biparietal diameter, abdominal circumference, femur length, estimated fetal weight, and umbilical artery Doppler) and uteroplacental data (uterine arteries Doppler velocimetry) were retrospectively collected from the routine second and third trimester ultrasound, as required by local guidelines. Delivery and pregnancy outcomes, including GWG at delivery, gestational age, mode of delivery, blood loss, birth weight, Apgar score, blood gas analysis, placental weight, fetal/placental weight ratio (FPR) (ratio between birth and placental weight as a proxy of placental efficiency), and newborn sex were recorded.

### 2.2. Statistical Analysis

All analyses were performed by using the statistical package SPSS, v.27 (IBM; Armonk, NY, USA). Continuous variables were expressed as median and range. Categorical variables were expressed as absolute and relative frequencies. Bivariate correlations were performed to investigate correlations between maternal baseline characteristics, FIGO nutritional score, hemodynamic parameters, and delivery outcomes. After performing a log-10 transformation to approximate the Gaussian distribution of non-normally distributed variables, multi-adjusted generalized linear models including confounding factors (maternal age, ethnicity, education, smoking habit, pregestational BMI, GWG at enrollment, conception mode, gestational age at enrollment, and fetal sex) were performed to evaluate associations between (1) maternal baseline characteristics (independent variables) and hemodynamic parameters (dependent variables); (2) FIGO nutritional score (independent variable) and hemodynamic parameters (dependent variables); (3) FIGO nutritional score and hemodynamic parameters (independent variables) and delivery outcomes (dependent variables: gestational age at delivery, GWG at delivery, birth weight, placental weight, fetal/placental weight ratio, blood loss, umbilical cord pH, and APGAR score). Statistical significance was considered at a *p*-value < 0.05.

## 3. Results

Of those recruited into the study, 143 low-risk singleton pregnancies were enrolled after excluding women with pregestational comorbidities (*n* = 6), language barriers (*n* = 3), and fetal anomalies (*n* = 2). Table 1 shows the baseline data and FIGO nutritional score. The study population, according to the eligible criteria, includes term pregnancies with physiological maternal characteristics. Nevertheless, of the total study population, 14.7% exhibited pregestational BMI in the range of overweight or obese (BMI > 24.9 kg/m^2^).

The single components of the FIGO nutritional score at enrollment are shown in Table 2. The major areas of nutritional deficiencies are related to sun exposure and the consumption of whole grains, sweets, and snacks.

Maternal hemodynamic parameters are presented in Table 3. The median values of hemodynamic variables were within the normal range.

Maternal pregestational BMI exhibited a negative correlation with CO (r = −0.29, *p* < 0.001) and SMII (r = −0.26, *p* < 0.01) and a positive correlation with SVR (r = 0.27, *p* < 0.001). No other maternal characteristics were correlated with hemodynamic parameters. The FIGO nutritional score was positively correlated with the inotropism index SMII (r = 0.27, *p* = 0.01), whereas no correlations were detected with CO and SVR. No correlations were found between hemodynamic parameters and second and third trimester fetal growth ultrasound data. Delivery outcomes are presented in Table 4 (missing data: *n* = 4). Due to the inclusion criteria and gestational age at enrollment, all women delivered at term with median placental and birth weights within the normal range. No adverse significant delivery outcomes were reported in our study population. Placental weight was positively correlated with CO (r = 0.241, *p* < 0.05) and negatively correlated with SVR (r = −0.21, *p* < 0.05). Furthermore, birth weight centile was positively correlated with CO (r = 0.26, *p* < 0.01) and negatively correlated with SVR (r = −0.25, *p* < 0.01). No correlations were found between inotropic index and birth outcomes.

Table 5 shows the results from the fully adjusted generalized linear models, including the previously defined confounding factors. Firstly, we investigated associations between baseline maternal characteristic and hemodynamic parameters. These models showed no associations between maternal hemodynamic parameters and maternal age, education, smoking habit, parity, GWG, and conception mode. Conversely, significant associations were confirmed between pregestational BMI and CO, SV, SVR, and SMII. Secondly, the same multi-adjusted models were used to investigate associations between the FIGO nutritional score and hemodynamic parameters. The FIGO nutritional score was positively associated with SMII independently of BMI, whereas no associations were detected with other hemodynamic parameters. Finally, the associations between FIGO nutritional score and hemodynamic parameters (independent variables) and delivery outcomes (dependent variables) were investigated. No associations were detected between the FIGO nutritional score and delivery outcomes. Conversely, CO was positively associated with birth and placental weights. On the other hand, SVR showed negative associations with birth and placental weights. No associations were detected between hemodynamic parameters and FPR or other delivery outcomes.

## 4. Discussion

### 4.1. Main Findigs

The present study investigated the associations between maternal characteristics, nutritional habits, hemodynamic parameters, and delivery outcomes among low-risk pregnancies at term. The results showed significant negative associations between maternal pregestational BMI and hemodynamic parameters in both univariate and multi-adjusted models, as highlighted by higher SVR and lower CO and SMII at increasing BMI values. No other maternal and pregnancy characteristics, including age, ethnicity, parity, education, smoke habit, conception mode, fetal sex or GWG, were associated with maternal hemodynamic parameters at term. Additionally, the adherence to a healthy diet and lifestyle as assessed by the FIGO nutritional score was associated with improved cardiac inotropism measured by SMII. Finally, maternal hemodynamic maladaptive parameters (higher SVR and lower CO) were associated with lower birth and placental weights despite being in the normal range.

The present study included low-risk singleton pregnancies with no known pregestational comorbidity, pregnancy complication, or fetal abnormality. Nevertheless, 14.7% of the total population had an inadequate nutritional status, being classified as overweight (12.6%) or obese (2.1%); 98.7% reported at least one nutritional risk defined by at least one negative answer in the FIGO nutritional checklist; and 19% provided five or more negative answers. These results highlight the relevant nutritional risk of an apparently low-risk pregnant population at term. Currently, we are in an obesogenic environment characterized by a widespread availability of low-quality and low-cost foods. This has led to excessive macronutrient intake, coupled with concomitant micronutrient deficiency, resulting in an increased risk of malnutrition. These alterations in micro and macronutrient intake can have significant consequences for both the mother and offspring.

Due to the limited sample size, a multivariate analysis stratified for BMI subgroups was underpowered.

Our findings about the association between BMI and hemodynamic maladaptations are consistent with previous data and suggest that pregestational BMI plays a primary role, independently of GWG, in maternal hemodynamic adaptations at term. In fact, in line with our data, Vinayagam et al. confirmed that obese pregnant women showed compromised cardiac function, reporting higher SVRI, MAP, and HR, and lower CI compared to lean controls [20,37,38].

Several explanations may be proposed. Firstly, the increased body fat of obese pregnancies is linked to increased circulating pro-inflammatory cytokine levels and oxidative stress, leading to chronic low-grade systemic inflammation, endothelial dysfunction, and peripheral vasoconstriction. Consequently, the heart is compelled to work against elevated pressure, possibly resulting in a decreased CO. Secondly, the renin–angiotensin–aldosterone system (RAAS) is more active in obese pregnancies, leading to increased renal tubular reabsorption. Consequently, this results in an expansion of blood volume, leading to increased venous return, SV, CO, and blood pressure. These changes result to be more significant during the third trimester of pregnancy [39]. Thirdly, obesity is linked to metabolic and endocrine derangements involving hyperlipidemia, hyperinsulinemia, hyperleptinemia, and estrogen concentration changes. Particularly, low density lipoproteins (LDL) can reduce extravillous cytotrophoblast migration and stimulate trophoblast apoptosis [40]. Insulin potentially reduces Placental Growth Factor (PlGF) synthesis and uterine trophoblast invasion, while leptin curtails cytotrophoblast proliferation and reduces PlGF concentrations [40]. All these factors may contribute to placental ischemia, and a reduction in angiogenic and increase in anti-angiogenic placental factors, as previously detected in obese pregnancies [41]. These factors may play a critical role in maternal vascular adaptations to pregnancy, leading to systemic endothelial dysfunction, reduced levels of vasodilator factors (i.e., nitric oxide and prostaglandins), and increased levels of vasoconstrictors and reactive oxygen species, resulting in increased SVR [40].

In line with previous findings, we did not detect associations between maternal parity, mode of conception, ethnicity, and hemodynamic parameters. Conversely, a predictable significant association between smoking habit and an increase in SVR was previously reported, which we did not confirm, possibly due to the reduced number of women who smoked during pregnancy (*n* = 4) [10].

In our study, we found a positive association between the FIGO nutritional score and cardiac inotropism in a multi-adjusted model. In other words, the adherence to a healthy diet and lifestyle improves the ability of heart contraction, independently of pregestational BMI, GWG, or other maternal characteristics. A possible explanation includes that an inadequate intake of micronutrients (with a key role played by magnesium, calcium, and potassium), as well as excessive consumption of sugars, can impair both cardiac function and vascular integrity [42,43]. Indeed, an adequate concentration of intracellular electrolytes is essential for maintaining the electrical balance in the heart, which is necessary to maintain effective cardiac contraction, and regulate vascular reactivity. Although the FIGO nutritional score does not provide quantitative intakes of each micronutrient, the single included items point to specific nutritional deficiencies. In particular, only 51.7% of our population showed regular sun exposure, thus highlighting a vitamin D inadequate status possibly affecting calcium metabolism and cardiac inotropism. Additionally, only 56.5% of our pregnancies regularly consume whole cereals, which contain vitamin B complex, playing a pivotal role in energy metabolism, and, indirectly, in cardiac inotropism.

No association was found between the FIGO nutritional score and delivery outcomes, contrary to what we previously reported among low-risk pregnancies enrolled in the first trimester [30]. This is likely due to differences in nutritional exposures realized in early pregnancy versus later.

Finally, we reported significant associations between increased SVR and lower CO and reduced fetal and placental growth, in line with previous reports [12,44,45]. Hemodynamic adaptations during pregnancy have been extensively studied in relation to FGR and hypertensive disorder (i.e., gestational hypertension and preeclampsia). Two potential explanations were proposed: some women exhibit high vascular resistance even prior to pregnancy, suggesting an intrinsic characteristic that could limit cardiovascular adaptation during pregnancy. On the other hand, FGR and hypertensive disorder seem to be linked to poor trophoblastic invasion of the maternal spiral arteries, resulting in the abnormal spiral remodeling of uterine arteries. This leads to a placental dysregulated system characterized by high velocity flow and elevated vascular resistances. Consequently, impedance in uterine arteries increases, contributing to elevated maternal vascular resistances and subsequently reducing CO. These hypotheses likely coexist in reality [14,44]. Our data show that higher SVR and lower CO are associated with lower birth and placental weights, despite being in the normal range, even in low-risk pregnancies. Partial alterations in the remodeling of uterine spiral arteries might cause a mild, non-pathological increase in SVR. In both cases, the alterations do not lead to clinically significant growth restriction, but rather to a “compensated” reduction in fetal and placental weights.

### 4.2. Study Strengths and Limitations

The strength of this study lies in its novel data, which assess the association between maternal characteristics, nutritional status and habits, hemodynamic parameters, and pregnancy outcomes. To our knowledge, this is the first study reporting the association between maternal nutritional habits and hemodynamic parameters in pregnancy, although the association between a Western-type dietary pattern and cardiovascular risk profile is well known in adult and elder populations [46,47,48].

Moreover, the study encompasses a well-defined cohort of healthy women with low-risk pregnancies, thereby reducing the potential biases associated with hemodynamic parameters.

Including both underweight and obese women enhances the external validity of our findings, aligning with the reported national prevalence of abnormal BMI among fertile women.

The nutritional checklist, developed and based on the FIGO recommendations concerning adolescent, preconception, and maternal nutrition, serves as a valuable tool for efficiently gathering information on dietary habits during pregnancy within clinical settings.

Although residual confounding could not be completely excluded, full adjustment for all known factors associated with hemodynamic parameters and pregnancy outcome was considered in the final model.

The most important limitation of this study is the lack of longitudinal observations of hemodynamic adaptations, extending from the first trimester, through pregnancy and up to delivery. Another significant limitation is the small sample size of the enrolled population. Finally, due to the use of the FIGO nutritional score, specific macro- and micronutrient intakes cannot be defined and quantified, thus making the interpretation of the effect on hemodynamic parameters or outcome only speculative.

## 5. Conclusions

Our study suggests that pregestational BMI is associated with a poorer hemodynamic profile at term, independently of GWG. Moreover, a well-balanced diet measured by the FIGO nutritional score is associated with improved cardiac inotropism in low-risk pregnancies. Therefore, the early optimization of BMI, dietary habits, and lifestyle during periconceptional period and pregnancy appears pivotal in order to improve maternal hemodynamic adaptations and pregnancy outcome. Furthermore, we observed that slight deviations in hemodynamic parameters are linked to lower birth and placental weights in a low-risk population. It is probable that the hemodynamic factors could further deteriorate in high-risk pregnancies, leading to the development of significant pregnancy complications. A nutritional screening of dietary habits through a reproducible, fast, and easy-to-check checklist seems to represent a useful tool for all healthcare providers, together with pregestational BMI, for ameliorating maternal hemodynamics, and finally pregnancy outcome. Further research, including intervention trials, is strongly needed.

## Figures and Tables

**Table 1 nutrients-16-00183-t001:** Maternal baseline characteristics and FIGO nutritional score values at enrollment in the total study population.

Maternal Characteristics		Total Study Population (*n* = 143)
Maternal age, years, median (range)		33 (20–43)
Maternal pregestational BMI, kg/m^2^, median (range)		21.0 (16.6–38.5)
Pregestational BMI, kg/m^2^, n (%)	18.5–24.9	110 (76.9)
	25–29.9	18 (12.6)
	≥30	3 (2.1)
GWG at enrollment, kg, median (range)		12 (3–24)
Gestational age at enrollment, days, median (range)		270 (254–286)
Educational level, n (%)	low	6 (4.2)
	intermediate	33 (23.1)
	high	100 (69.9)
Working status, n (%)	unemployed	19 (13.3)
	employed	118 (82.5)
Marital status, n (%)	not married	62 (43.4)
	married	77 (53.8)
Ethnicity, n (%)	non-Caucasian	19 (13.3)
	Caucasian	124 (86.7)
Conception mode, n (%)	spontaneous	137 (95.8%)
	ART	6 (4.2%)
Smoking habit, yes n (%)	yes	5 (3.5)
Nulliparous, n (%)		95 (66.4)
FIGO nutritional score, median (range)		7 (3–10)

Continuous variables are expressed as medians and ranges. Categorical variables are expressed as absolute and relative frequencies. BMI: body mass index; GWG: gestational weight gain; ART: assisted reproductive technology; FIGO: International Federation of Gynecology and Obstetrics.

**Table 2 nutrients-16-00183-t002:** Single items of the FIGO nutritional score as a measure of maternal adherence to a healthy diet and lifestyle.

Score Component	Frequency (*n* = 143)
Meat, yes n (%)	113 (79.0)
Fruit and vegetables, yes n (%)	113 (79.0)
Fish, yes n (%)	112 (78.3)
Dairy, yes n (%)	113 (79.0)
Whole cereals, yes n (%)	81 (56.6)
Sweets and snacks, yes n (%)	95 (66.4)
Hemoglobin, yes n (%)	112 (78.3)
Folic acid, yes n (%)	133 (93.0)
Iodized salt, yes n (%)	122 (85.3)
Sun exposure, yes n (%)	74 (51.7)

Values are expressed as absolute and relative frequencies.

**Table 3 nutrients-16-00183-t003:** Maternal hemodynamic parameters in the total study population.

Hemodynamic Parameters	Total Study Population (*n* = 143)
BSA, median (range), m^2^	1.8 (1.4–2.3)
HR, median (range), bpm	77.7 (44.7–123.4)
SV, median (range), cm^3^	44.0 (25.0–68.8)
CO, median (range), L/min	6.4 (3.4–9.6)
MAP, median (range), mmHg	80 (60–100)
SVR, median (range), dyne s cm^−5^	1000.0 (609.4–1651.4)
SMII, median (range), Watt/m^2^	1.6 (0.9–2.6)

Values are expressed as median and range. BSA: body surface area; HR: heart rate; SV: stroke volume; CO: cardiac output; MAP: mean arterial pressure; SVR: systemic vascular resistance; and SMII: Smith–Madigan inotropy index.

**Table 4 nutrients-16-00183-t004:** Delivery outcomes of the study population.

Delivery Outcomes		Total Study Population (*n* = 139)
GWG at delivery, kg, median (range)		12.0 (4.0–25.0)
Gestational age at delivery, days, median (range)		281 (267–292)
Delivery mode, n (%)	Vaginal delivery	105 (75.5)
	Vacuum-assisted operative delivery	10 (7.2)
	Caesarean section	24 (17.3)
Birthweight, g, median (range)		3405 (2500–4630)
Birthweight, centile, median (range)		51 (4–99)
Blood loss, mL, median (range)		400 (100–1550)
Placental weight, g, median (range)		550 (350–890)
FPR, median (range)		6.1 (4.5–10.1)
pH, median (range)		7.24 (6.98–7.53)
BE, median (range)		−5.0 (−16.3–4.2)
Lactate, median (range)		4.0 (1.2–12.9)
Apgar-1, median (range)		9 (6–10)
Apgar-5, median (range)		10 (8–10)
Neonatal sex, n (%)	Male	71 (51)
	Female	68 (49)

Continuous variables are expressed as median and ranges. Categorical variables are expressed as absolute and relative frequencies. GWG: gestational weight gain; FPR: fetal placental weight ratio; BE: base excess.

**Table 5 nutrients-16-00183-t005:** Results from the multi-adjusted generalized linear models to evaluate associations between maternal baseline characteristics and hemodynamic parameters; FIGO nutritional score and hemodynamic parameters; hemodynamic parameters and delivery outcomes.

Independent Variable	Dependent Variable	β (95%CI)	*p*-Value
BMI	CO	−0.07 (−0.12; −0.03)	<0.001
	SV	−0.92 (−1,47; −0.37)	0.001
	SVR	43.16 (17.36; 68.96)	<0.001
	SMII	−0.02 (−0.04; −0.00)	0.02
Nutritional score	SMII	0.05 (0.02; 0.09)	0.005
CO	Neonatal weight	7.48 (0.01; 14.95)	0.05
	Placental weight	44.27 (13.77; 74.77)	<0.01
SVR	Neonatal weight	−0.01 (−0.03; −0.01)	0.02
	Placental weight	−0.06 (−0.11; −0.01)	0.02

All models include adjustment for maternal age, ethnicity, education, smoking habit, pregestational BMI, GWG and gestational age at enrolment, conception mode, and fetal sex. Statistical significance *p* < 0.05. BMI: body mass index; CI: confidence interval; CO: cardiac output; SV: stroke volume; SVR: systemic vascular resistance; SMII: Smith–Madigan inotropy index; GWG: gestational weight gain.

## Data Availability

All data that support the findings of this study are available from the authors C.L. and F.P. on reasonable request. The data are not publicly available due to privacy.

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
