# Peer review of "Associations between Maternal Nutritional Status, Hemodynamic Parameters, and Delivery Outcomes in Low-Risk Pregnancies: A Prospective Observational Study"

_nutrients, 2024, doi:10.3390/nu16020183_

Round 1
Reviewer 1 Report
Comments and Suggestions for Authors
Over half (24 out of 43) of the references are older than 5 years.
Add a glossary of abbreviations.
Line 75 - remove the quotes from Medical Ethical ......
Line 93 - remove the 'a' before 1-point
Line 111 - Add the ) after USA
Line 117 - add an 'l' to enrollment
Line 117 - what is conception mode?
Line 127 - Do not start the sentence with a number. You might put "Of those recruited into the study, 143 low-risk....
Line 129 - Add "of the study population, 14.7%....."
Add the normal BMI to Table 1. Also, change the word "worker" to working outside the home or employed.
Line 137 - Delete Table 2 shows and replace with "The single components of the FIGO nutritional score at enrollment are shown in Table 2.
Line 156 - if there are 4 data points missing the n=139. Please explain the discrepancy in Table 4 where it says n=140. Table 3 says n=143.
In the Delivery mode the numbers do not add up 88+10+20 is NOT 140. Correct this and explain operative vaginal delivery.
In the neonatal sex category, it should equal 100% but does not. 50% +48% is 98%. Is the sex of the other 2% unknown?
LIne 202 - add "being" after despite.
Line 208 - delete "even" before five
Line 229 - delete "finally"
Line 223 - put a comma after confirm
Line 245 Delete "Despite" and replace with "Although"
Line 251 - change sentence to "... exposures realized in early recruitment versus later entry into the study" or something similar.
Line 263 - add "being" after despite
Comments on the Quality of English Language
See above suggestions.
Reviewer 2 Report
Comments and Suggestions for Authors
This is a simple cum-hoc research design: pregnant women at the end of pregnancy (always a sure supply), some easy-to-fill-out one-time questionnaire, the research diagnostic technique (USCOM), and the sure-to-follow-right-in-my-lap birth data. Then some statistical software to unleash, and there we are.
Unsurprisingly, the study offers little novelty. The limitations summed up in a quick paragraph at the end of the report (lines 288-293) are pertinent: no longitudinal data, and a rough measure of nutritional quality. The relationship between BMI/nutritional score and maternal hemodynamics appears to be resilient, popping up in a one-shot cum-hoc design.
Regarding the nutritional quality assessment, it is established knowledge that eating lots of barbecued red meat and pre-packaged red meat is quite different from the spare use of cooked lean white meat; any dietitician will confirm that sugary fruits (grapes, etc.) are best avoided in pregnancy (GDM patients are forbidden to use them!), whereas vegetables are generally healthy; zero-fat and high-fat dairy are not quite the same, etc.
Line 247-9: "only 51.7% ... showed regular sun exposure, thus highlighting a vitamin D inadequate status". My guess is that people understood regular sun exposure as sunbathing. The study was carried out between May and August, sunny months in Italy to my knowledge. It is generally stated that a 10 min exposure of both arms in summertime is enough to avoid vitamin D deficiency. Hence, without more detailed questionnaire data or measuring 25(OH)D levels, the data in my view say nothing about the vitamin D status of the study participants.
